# Absence of causality between seismic activity and global warming

Mikhail Y. Verbitsky[1,2], Michael E. Mann[3], and Dmitry Volobuev[4]

[1]Gen5 Group, LLC, Newton, MA, USA
[2]UCLouvain, Earth and Life Institute, Louvain-la-Neuve, Belgium
[3]University of Pennsylvania, Department of Earth and Environmental Science, Philadelphia, PA, USA
[4]The Central Astronomical Observatory of the Russian Academy of Sciences at Pulkovo, Saint Petersburg, Russia

Correspondence: Mikhaïl Y. Verbitsky (verbitskys@gmail.com)

**Abstract.** There is no more consequential scientific matter today than global warming. The societal and policy implications, however, hinge upon the attribution of that warming to human activity, and specifically, continued societal reliance on the burning of fossil fuels. It was recently suggested that this warming can be explained by the non-anthropogenic factor of seismic activity. If that were the case, it would have profound implications. We have accessed the validity of the claim using a statistical technique (the method of conditional dispersion) that evaluates the existence of causal connections between variables, finding no evidence for any causal relationship between seismic activity and global warming.

The anthropogenic cause of planetary warming during the industrial era is well established (e.g., Stocker et al, 2014). That does not mean, however, that alternative hypotheses challenging an anthropogenic cause of observed warming shouldn't be evaluated on their merit. It has been recently proposed that the warming (particularly in polar regions) can be attributed to tectonic waves caused by large earthquakes and by the subsequent destruction of the microstructure of gas hydrates and release of the methane (Lobkovsky et al, 2022). To test this hypothesis, we apply the method of conditional dispersion (Čenys et al., 1991, Verbitsky et al, 2019) to explore a potential causal relationship between temperature and global seismic activity. The method has been proved to be less noise-sensitive than convergent-cross-mapping algorithms and more universal than prediction-improvement approaches. Briefly, the method assumes that if two variables are dependent (or in other words, the causality in Wiener's definition exists), then they belong to the same dynamical system and therefore if points of the first variable (e.g., seismicity index) are close, the synchronous points of the second variable (e.g., temperature) should also be close. Thus, the dependence of the conditional dispersion $\sigma(\varepsilon)$ of the temperature variable upon the distance $\varepsilon$ between synchronous points of the seismic-activity variable becomes a signature of causal relationship between the temperature and the seismic activity. Specifically, if the seismic activity is the cause of warming, then the conditional dispersion $\sigma(\varepsilon)$ of the temperature variable should decrease when the distance $\varepsilon$ between synchronous points of the seismic-activity variable decreases.

In Figure 1 we present the results of the conditional dispersion calculations together with the data. Specifically, we use the earthquake magnitude data (Ammon et al, 2010) supplemented by the most recent fragment from the IRIS DMC database (https://ds.iris.edu/wilber3/find_event). We defined the seismicity index as expected maximum values of crustal deformation, described by the empirical law of Okada (1995), i.e., $\lg(U_{max}) = 1.5M - 2lgR - 6.0$, where $U_{max}$ (cm) are expected maximum values of crustal deformation, $M$ is earthquake magnitude, and $R$ (km) is hypocentral distance to the region of interest. Based on this law, we created three seismicity indexes: (a) in the first one, only the earthquake magnitude $M$ is taking into account, and the hypocentral distance $R$ is used only as a scaling constant; (b) in the second index, both earthquake magnitude $M$ and the hypocentral distance $R$ to the North Pole are accounted for, and (c) the third index accounts for both earthquake magnitude $M$ and the hypocentral distance $R$ to the South Pole. The Global Land-Ocean Temperature Index (Hansen et al, 2010, Lenssen et al, 2019, https://data.giss.nasa.gov/gistemp/) has been used as the global temperature data $T_{Gl}$.

It can be seen that for all three seismicity indexes, the conditional dispersion of global temperature
anomalies $\sigma(\varepsilon)$ is independent of $\varepsilon$ where $\varepsilon$ is the distance between synchronous points of a seismicity
index. In other words, ***there is no causal relationship between seismic activity and global warming***. For
comparison, we show in Figure 1 the conditional dispersions of global temperature anomalies where $\varepsilon$ is
the distance between synchronous points of atmospheric $CO_2$ concentration. The causality between
atmospheric carbon dioxide concentration and temperature anomalies, by contrast, is clear.
In conclusion, there is no statistical support for the proposition that seismic activity is a cause of
large-scale warming in recent decades. A parallel analysis of $CO_2$ and temperature supports the prevailing
hypothesis that this warming is substantially caused by an increase in greenhouse gas concentrations from
fossil fuel burning.
**Code availability**
The MATLAB R2021b code and data to reproduce the results as they are presented in Fig. 1 are available
at https://zenodo.org/records/11233609 (Volobuev, 2024).
**Data availability**
This paper refers exclusively to published research articles and their data. We refer the reader to the cited
literature for access to data.
**Author contributions**
MYV conceived the research, DV performed the computations. MYV, MEM, and DV jointly discussed
the findings and contributed equally to writing of the manuscript.
**Competing interests**
The contact author has declared that the authors have no conflict of interest.
**Acknowledgements**
We are grateful to our two anonymous reviewers for their helpful suggestions.

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

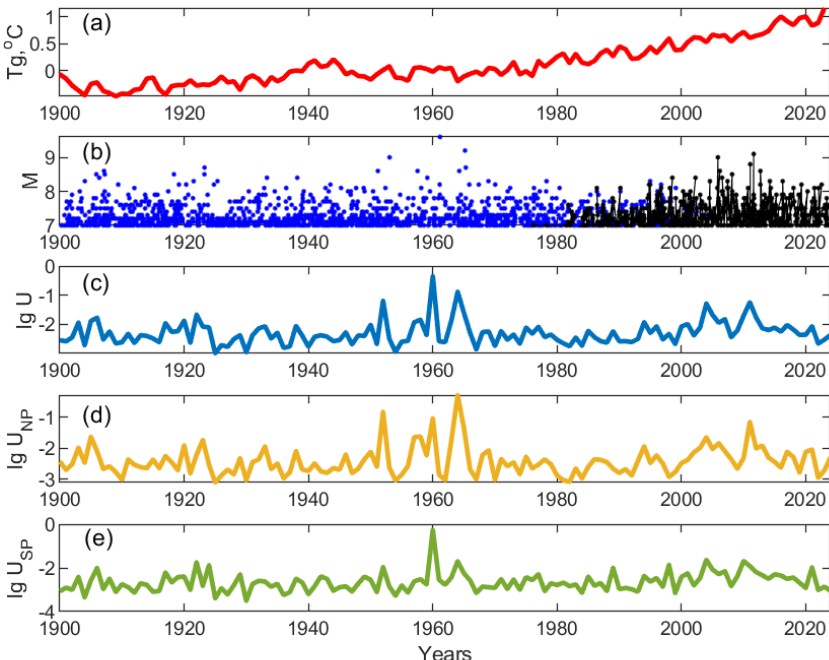




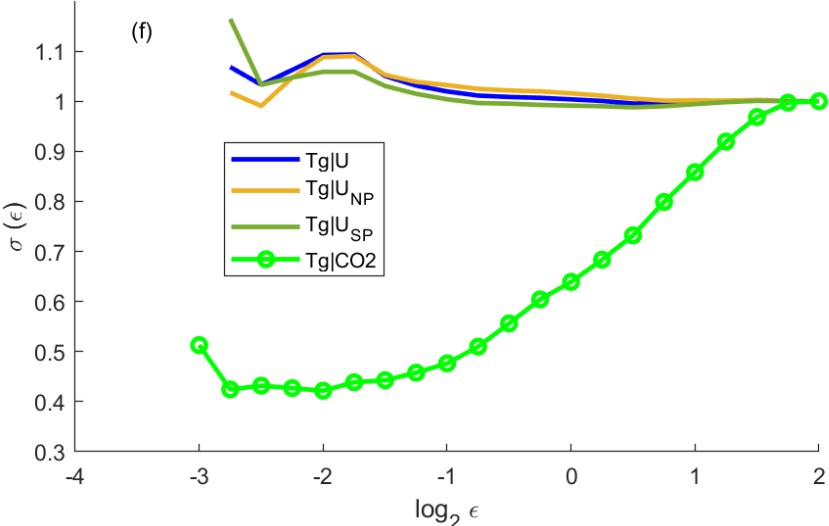

**Figure 1.** (a) Global temperature anomalies data; (b) Earthquake magnitude data of Ammon et al (2010),
**dark blue**, supplemented by the most recent fragment from the IRIS DMC database
(https://ds.iris.edu/wilber3/find_event, **black**; (c) The seismicity index with only earthquake magnitudes
$M$ taken into account, **blue**; (d) The seismicity index where earthquake magnitudes $M$ and the hypocentral
distance $R$ to the North Pole are accounted for, **yellow**; (e) The seismicity index where earthquake
magnitudes $M$ and the hypocentral distance $R$ to the South Pole are accounted for, **dark green**; (f)
Conditional dispersions of global temperature anomalies $\sigma(\varepsilon)$, where $\varepsilon$ is the distance between
synchronous points of a seismicity index of a corresponding color and conditional dispersion of global
temperature anomalies $\sigma(\varepsilon)$, where $\varepsilon$ is the distance between synchronous points of atmospheric $CO_2$
concentration, **green circled line**.