# Peer review of "Absence of causality between seismic activity and global warming"

_Earth System Dynamics, 2024_

## Author Response (AR1)

Dear Prof. Didenkulova,

We are sincerely grateful to you and to our anonymous reviewers. We adopted all their comments and hope that the manuscript is now close to their expectations. All changes are marked red.

Respectfully,

Mikhail Verbitsky, Michael Mann, and Dmitry Volobuev

**Response to Anonymous Referee #1**

Dear Anonymous Referee #1,

We are grateful for your insightful review that will help us to improve the manuscript. Your *verbatim* comments are below (in bold), each followed by our response.

**General comment: The paper aims at showing the non-existence of correlation between seismic activity and climate variability. The paper is very short, simple and easy to read.**

**Response:** We are pleased that you find our paper being simple and easy to read.

**I have a few comments about this paper that should be addressed in a revised version.**

**Comment 1: The authors use an unconventional tool (to my knowledge this is used only for climate researches) to check the correlation between seismicity and climate. I am wondering why do not use more classical statistical tools that have a solid theoretical mathematical basis. In essence, the authors should justify better their choice.**

**Response:** Our paper is not a regular research article but **ESD Letter** that is designed to **briefly** report important results and therefore it is limited to 2,500 words. For this reason, we didn't provide the full substantiation of the method of conditional dispersion but referenced our previous work where such substantiation has been performed (Verbitsky, M. Y., Mann, M. E., Steinman, B. A., and Volobuev, D. M.: Detecting causality signal in instrumental measurements and climate model simulations: global warming case study, Geosci. Model Dev., 12, 4053–4060, https://doi.org/10.5194/gmd-12-4053-2019, 2019).

At the same time, we agree with you that some readers may benefit from better understanding of our instrument choice. Since all discussions, including this one, are going to be published as supplemental materials, the long quote from Verbitsky et al (2019) may be helpful for devoted readers:

"The most simplistic approach, the Pearson correlation between two time series, which is often mentioned in the context of causality, does not really measure the causality. While statistically significant correlation quantifies similarity between time series, it does not imply a causality resulting from physical relationships between the natural processes that are expressed by the time series and that can be modeled using differential equations. Instead, it provides a statistical test of a hypothesis that describes a physical link between the two variables (i.e., expressed as time series) without actually testing either the direction of causality or the plausibility of the physics underlying the hypothesis. The breakthrough Granger developments (Granger, 1969) provided a foundation for several causality-measuring techniques based on different hypotheses of data origin. The requirement of the cause leading the effect (but not vice versa) defines the direction of a causal link if a more general hypothesis of lagged linear connection between noisy autoregressive processes is assumed. Though this hypothesis leads to statistically significant estimates of climate response to the forcing input (e.g., Kaufmann et al., 2006, 2011; Attanasio, 2012; Attanasio et al., 2012; Mokhov et al., 2012; Triacca et al., 2013), it may not be able to reliably detect the direction of causality in the climate system because the potential for non-linearities in the climate system (leading to extreme sensitivity to initial conditions, i.e., deterministic chaos) is not taken into account. For example, Paluš et al. (2018) demonstrated that coupled chaotic dynamical systems can "violate the first principle of Granger causality that the cause precedes the effect." The Shannon information flow approach expands Granger causality to non-linear systems, using transfer entropy as a causality measure. Barnett et al. (2009) have shown that transfer entropy is equivalent to Granger causality for Gaussian processes. The transfer entropy between two probability distributions is typically considered the most general approach for causality detection, and numerous modifications of transfer-entropy-based causality-measuring techniques have been developed for different applications (Pearl, 2009), including causality measurements of global warming (e.g., Stips et al., 2016). It should be noted though that all probability-based causality measures require long time series to calculate statistical distributions and may lack applicability to local climate due to high inhomogeneity and non-stationarity of the data (e.g., O'Brien et al., 2019). The prediction improvement approach is often considered as a generalization of Granger causality for non-linear systems (e.g., Krakovská and Hanzely, 2016). It is highly practical and, besides causality calculations, it may help to improve the prediction accuracy. For pure causality purposes, however, it adds an additional uncertainty because the causality may depend on the chosen prediction method. The convergent cross-mapping approach (Sugihara et al., 2012; Van Nes et al., 2015) has been recently designed to work with relatively short data series, thus addressing the major constraint of transfer-entropy approach. The background hypotheses of the method is more narrow and includes only non-linear dynamical systems, though convergent cross mapping remains applicable to most natural systems in ecology and geosciences (Sugihara et al., 2012). The approach considers conditional evolution of nearest neighbors in the reconstructed Takens' space, so it is sensitive to the noise and may not be applicable to a wide range of timescales. Moreover, Paluš et al. (2018) have shown that convergent cross mapping is not capable of determining the directionality of a causal link. Therefore, identification of specific causal effect measures for climate observables is still a challenge. When causal effect measures are identified, the graph theory could be employed for further analysis of multiple causality chains (Hannart et al., 2016; Runge et al., 2015). Along with dimensionality reduction formalism (e.g., Vejmelka et al., 2015), it may lead to a promising general approach.   For our case study, we advocate the method of conditional dispersion (MCD) developed by Čenys et al. (1991) as a causal effect measure. It has also been designed for non-linear systems and exploits the asymmetry of the conditional dispersion of two variables in Takens' space along all available scales. Therefore, it remains more general and noise resistant than convergent cross-mapping techniques and more general than prediction improvement approaches because it is insensitive to the choice of the prediction method."

Interestingly, in 2019, we concluded our paper with the following: "With our calculations, we calibrate MCD against existing measurements and simulations. As long as MCD is trusted as an insightful approach, it can be used for express testing of new models and, perhaps more importantly, *can serve as a first test for any new external forcing candidate that may be considered as an alternative or supplement to $CO_2$.*" This is exactly where we are 5 years later: A new forcing candidate has been proposed, and we use method of conditional dispersion as the first test.

**Action:** We will add (within ESD Letters limit) a few sentences to better justify our choice.

**Done: New lines 26-28**

**Comment 2: As an additional comment, the authors use the count of earthquakes above M7. This is hardly justifiable, because a M8 releases 32 times the energy of a M7, and a M9 releases about 1000 times more energy than a M7. So, close M7 events do not have the same energy than a single M8 or larger. In essence, I do not think that the number of M7+ is a good proxy to measure the seismicity index. Maybe considering the seismic energy could be more appropriate. Even better, the seismicity index could be quantified by the ground shaking at polar regions (or any other region of interest)**

**produced by any single earthquake of the catalog used. This is feasible, but it would likely require an additional work made by a seismologist.**

**Related to the previous point, if the interest is on the polar region, the seismicity index should be weighted according to the location of earthquakes, since earthquakes close to polar regions may produce stronger shaking in these regions, with respect to larger earthquakes that occur on equatorial regions. This is made simulating the ground shaking of each earthquake (seismic waves decreases with distance from the hypocenter), or, in case the authors use the energy released by each single earthquake, weighting the distance of earthquakes to polar regions with an appropriate spatial decay (roughly speaking, the surface waves decay with 1/r only due to the geometrical spreading).**

**Response:** We find your suggestion to be very valuable. Therefore, we have redefined the seismicity index as expected maximum values of crustal deformation and recalculated it according to the empirical law of Okada (1995) (Okada, Y.: "Simulated empirical law of coseismic crustal deformation." Journal of Physics of the Earth 43, 6, 1995, 697-713):

$$\lg(U_{max}) = 1.5M - 2lgR - 6.0$$

Here $U_{max}$ are expected maximum values of crustal deformation (cm), $M$ is earthquake magnitude, and $R$ is hypocentral distance to the region of interest.

Based on this law, we created three seismicity indexes: (a) in the first one, only the earthquake magnitude $M$ is taking into account, and the hypocentral distance $R$ is used only as a scaling constant; (b) in the second index, both earthquake magnitude $M$ and the hypocentral distance $R$ to the North Pole are accounted for, and (c) the third index accounts for both earthquake magnitude $M$ and the hypocentral distance $R$ to the South Pole.

Results of the new seismicity-indexes and corresponding causality calculations are presented in Figure 1. It can be observed that, in all three cases, our results did not change: The conditional dispersion of global temperature anomalies σ(ε) is independent of ε where ε is the distance between synchronous points of a seismicity index. In other words, there is no causal relationship between seismic activity and global warming.

**Action:** We believe that the seismicity indexes, recalculated per your recommendations, make our results more robust and we will update our ESD Letter accordingly.

**Done: New lines 39-47, 49-51, 89-90, 102-111, and new Figure 1.**

[Figure]

**Figure 1.** Panels from top to bottom: Global temperature anomalies data (**red**); Earthquake magnitudes (**blue and black**); The seismicity index with only earthquake magnitudes *M* taken into account (**blue**); The seismicity index where earthquake magnitudes *M* and the hypocentral distance *R* to the North Pole are accounted for (**yellow**), The seismicity index where earthquake magnitudes *M* and the hypocentral distance *R* to the South Pole are accounted for (**dark green**), Conditional dispersions of global temperature anomalies σ(ε), where ε is the distance between synchronous points of a seismicity index of a corresponding color and conditional dispersion of global temperature anomalies σ(ε), where ε is the distance between synchronous points of atmospheric $CO_2$ concentration (**green circled line**).

**Response to Anonymous Referee #2**

**Comment: The authors tried to prove the argument by Leopold I. Lobkovsky , "Trigger mechanisms of gas hydrate decomposition, methane emissions, and glacier breakups in polar regions as a result of tectonic wave deformation", is wrong. The paper was published in an open-access journal of MDPI, which does not have a good reputation. From google scholar, I saw that the paper has been cited for 16 times; however, half of them are from Leopold I. Lobkovsky. I think most of researcher would not pay any attentions on such kind of papers. In addition, I think that this MS is more like a comment on the paper of Leopold I. Lobkovsky rather than a formal academic paper**

**Response:**

Dear Anonymous Referee #2,

Thank you for your review. We understand that you do not have concerns regarding our method and conclusions, and we are grateful for that.

While we recognize that there is variable quality of the papers that appear in many journals, including MDPI journals, this is not reason for declaring them unworthy of comment or response simply for that cause, particularly when a novel—if seemingly unlikely—mechanism has been proposed and, as yet, remains untested or challenged, as is the case for the claimed relationship between seismic activity and warming. More detailed, overall supportive reviews by Referee #1 and Referee #3 do not challenge the merit of our analysis. Instead, they suggest some additional considerations to insure the robustness of our findings, which we look forward to incorporating in our revisions.

As far as the short nature of the contribution, we hasten to point out that it is not a regular paper but an ESD Letter, short contributions which are limited to no more than 2,500 words.

Thank you again,

Mikhail Verbitsky and Michael E. Mann

**Action:** No action is required

**Response to Anonymous Referee #3**

Dear Anonymous Referee #3,

We are grateful for your insightful review that will help us to improve the manuscript. Your *verbatim* comments are below (in bold), each followed by our response.

**The manuscript concisely elaborates on the viewpoint of "Absence of causality between seismic activity and global warming" with a very brief text. The curve in Figure 1 of the manuscript is highly persuasive.**

**Response:** We are pleased that you find our paper being concise and highly persuasive.

**However, I still have some doubts: Firstly, what was the consideration behind only counting earthquakes of magnitude M>=7 in the manuscript, and does this have any impact on the results? Secondly, how was the seismic activity index calculated in the manuscript?**

**Response:** Initially, in our preprint, the annual seismicity index was calculated simply as an annual number $N$ of earthquakes with magnitude $M \geq 7$ with the aim to account for most powerful earthquakes. Per Reviewer #1 recommendation, to better account for energy of earthquakes that is proportional to $10^{1.5M}$ such that "M8 releases 32 times the energy of a M7, and a M9 releases about 1000 times more energy than a M7", and also to account for geographical location of earthquakes, we have redefined the seismicity index as expected maximum values of crustal deformation and recalculated it according to the empirical law of Okada (1995) (Okada, Y.: "Simulated empirical law of coseismic crustal deformation." Journal of Physics of the Earth 43, 6, 1995, 697-713):

$$\lg(U_{max}) = 1.5M - 2lgR - 6.0$$

Here $U_{max}$ are expected maximum values of crustal deformation (cm), $M$ is earthquake magnitude, and $R$ is hypocentral distance to the region of interest.

Based on this law, we created three seismicity indexes: (a) in the first one, only the earthquake magnitude $M$ is taking into account, and the hypocentral distance $R$ is used only as a scaling constant; (b) in the second index, both earthquake magnitude $M$ and the hypocentral distance $R$ to the North Pole are accounted for, and (c) the third index accounts for both earthquake magnitude $M$ and the hypocentral distance $R$ to the South Pole.

Results of the new seismicity-indexes and corresponding causality calculations are presented in Figure 1. It can be observed that, in all three cases, our results did not change: The conditional dispersion of global temperature anomalies σ(ε) is independent of ε where ε is the distance between synchronous points of a seismicity index. In other words, there is no causal relationship between seismic activity and global warming.

**Action:** We believe that these new seismicity indexes make our results more robust and we will update our ESD Letter accordingly.

**Done: New lines 39-47, 49-51, 89-90, 102-111, and new Figure 1.**

[revised manuscript text omitted]